# ArtSymbioCyc, a metabolic network database collection dedicated to arthropod symbioses: a case study, the tripartite cooperation in *Sipha maydis*

Patrice Baa-Puyoulet,[1] Léo Gerlin,[1] Nicolas Parisot,[1] Sergio Peignier,[1] François Renoz,[2] Federica Calevro,[1] Hubert Charles[1]

**ABSTRACT**   Most arthropods live in close association with bacteria. The genomes of associated partners have co-evolved, creating situations of interdependence that are complex to decipher despite the availability of their complete sequences. We developed ArtSymbioCyc, a metabolism-oriented database collection gathering genomic resources for arthropods and their associated bacteria. ArtSymbioCyc uses the powerful tools of the BioCyc community to produce high-quality annotations and to analyze and compare metabolic networks on a genome-wide scale. We used ArtSymbioCyc to study the case of the tripartite symbiosis of the cereal aphid *Sipha maydis* focusing on amino acid and vitamin metabolisms, as these compounds are known to be important in this strictly phloemophagous insect. We showed that the metabolic pathways of the insect host and its two obligate bacterial associates are interdependent and specialized in the exploitation of Poaceae phloem, particularly for the biosynthesis of sulfur-containing amino acids and most vitamins. This demonstrates that ArtSymbioCyc does not only reveal the individual metabolic capacities of each partner and their respective contributions to the holobiont they constitute but also allows to predict the essential inputs that must come from host nutrition.

**IMPORTANCE**   The evolution has driven the emergence of complex arthropod-microbe symbiotic systems, whose metabolic integration is difficult to unravel. With its user-friendly interface, ArtSymbioCyc (https://artsymbiocyc.cycadsys.org) eases and speeds up the analysis of metabolic networks by enabling precise inference of compound exchanges between associated partners and helps unveil the adaptive potential of arthropods in contexts such as conservation or agricultural control.

**KEYWORDS**   symbiosis, metabolism, metabolic network, insects, database, Aphididae, *Buchnera aphidicola*, *Serratia symbiotica*, amino acids, vitamins

The holobiont and hologenome concepts offer a broader vision of the individual by including the microbial organisms associated with the host—whether external, internal, or even intracellular—into a single system (1–3). The pervasiveness, as well as the functional and evolutionary importance, of symbiotic relationships is now widely established and questions the very notions of individuals and species (4, 5). Indeed, we can no longer consider individuals as entities but must see the network of complex interactions that make up the fabric of life. Since individuals can be considered as interacting agents in a network of interactions, it seems just as important to understand the holobiont's overall as the intrinsic capacities of each partner (6).

This is particularly true for insects, which constitute the most diverse phylum of animals (7). They make an essential contribution to ecosystem services worldwide, including soil formation and pollination (8). They are vectors of numerous human,

**Peer Reviewer** Rosario Gil, Paterna, Valencia, Spain

Address correspondence to Patrice Baa-Puyoulet, patrice.baa-puyoulet@inrae.fr, or Hubert Charles, hubert.charles@insa-lyon.fr.

The authors declare no conflict of interest.

See the funding table on p. 13.

animal, and plant diseases, as well as pests of all major crops worldwide (9). In this sense, understanding their adaptation mechanisms, notably through the symbiotic relationships they maintain with microorganisms, is essential to decipher how insects can cope with environmental changes and improve control strategies against certain pests (10, 11). The ever-increasing number of sequencing projects allows for carrying out global genomic analyses for a very wide range of insect species. To date, NCBI's genomics section references thousands of insects and around two million bacteria, of which 516 genomes are associated with the keyword "endosymbionts," making it possible to reconstruct, at least in part, certain hologenomes consisting of the host and its associated endosymbiotic or gut bacteria.

Automated reconstruction of metabolic networks from the genomic information of holobiont partners remains a challenge, particularly with regard to the consistency of functional annotations on a genome scale. For that purpose, we previously built the CycADS information system (12), which aggregates genome-scale functional annotations from different methods with identical pipelines and parameter settings for all interacting organisms to ensure a consistent description of enzymes, reactions, and pathways and hence to guarantee the comparability of their respective networks. In the present work, we used CycADS in conjunction with the Pathway Tools software (13) to build the BioCyc-like (14) ArtSymbioCyc collection of databases (https://artsymbiocyc.cycad-sys.org/). During the automated and expert analysis process, we also exploited the Pathway Tools embedded MetaCyc database (15), a curated database of experimentally elucidated metabolic pathways from all domains of life, which facilitates researchers, in particular, to identify gaps in pathways and potential metabolic complementations between associated partners. Finally, the ArtSymbioCyc interface enables users to determine the precursors, intermediates, and final compounds that must necessarily be imported from food, transformed, and transferred between associated organisms.

ArtSymbioCyc contains the metabolic networks, based on the available genomic information, of 10 holobionts corresponding to six crop pest host-insects (*Acyrthosiphon pisum*, *Cinara cedri*, *Sipha maydis*, *Bemisia tabaci* strains MEAM1 and MED, and *Sitophilus oryzae*), three blood-eaters host-insects (*Pediculus humanus corporis*, *Cimex lectularius*, and *Glossina morsitans*), and the model insect *Drosophila melanogaster*. These hosts are accompanied by 1 to 4 of the 21 intracellular endosymbionts or commensal-associated bacteria listed in the collection (Table S1).

To illustrate the main feature usage of ArtSymbioCyc, we have decrypted the metabolic network of the *S. maydis* holobiont, composed here by the aphid host, that was recently sequenced (16), and its co-obligate symbionts, *Buchnera aphidicola* and *Serratia symbiotica* (17). The cereal aphid *S. maydis* (Chaitophorinae) feeds on many grass species (Poaceae) and is distributed worldwide in most temperate climates where it often damages cereal crops (18). Like all strict phloem-feeding insects, *S. maydis* is dependent on obligate symbionts, which supply it with compounds that are too rare or absent from its diet. Although housed in different bacteriocytes within the aphid body cavity, the two nutritional endosymbionts are spatially close. *S. symbiotica* is confined to large syncytial secondary bacteriocytes embedded between uninucleate primary bacteriocytes containing *B. aphidicola* (16, 17). This topology facilitates metabolic exchanges, and the two symbiotic bacteria carry out or participate in reactions allowing the synthesis of essential amino acids and vitamins (17).

ArtSymbioCyc has enabled us to depict the intertwined metabolisms of these three associated species and to finely track the origin and the fate of each metabolite to identify the main compounds that must feed the system. We propose the ArtSymbioCyc collection as a relevant and user-friendly resource to decipher the intimate and complex co-dependencies on which arthropod holobionts are based.

## RESULTS AND DISCUSSION

### ArtSymbioCyc, a collection of metabolic networks of arthropod symbioses

ArtSymbioCyc is a Pathway Genome DataBase (BioCyc PGDB) collection containing the metabolic networks inferred from the reference genomes of 10 insects accompanied by their respective intracellular symbionts or commensal bacteria: three aphid species *A. pisum*, *C. cedri*, and *S. maydis*; two strains of the sweet potato whitefly *B. tabaci*; the bed bug *C. lectularius*; the fruit fly *D. melanogaster*; the tsetse fly *G. morsitans*; the body louse *P. humanus corporis*; the rice weevil *S. oryzae*. The list of all the databases currently contained in ArtSymbioCyc as well as the description of the main features of the corresponding metabolic network are reported in Table 1 and Table S1.

The ArtSymbioCyc metabolic networks were reconstructed using CycADS (12), an annotation management system originally developed for the reconstruction of pea aphid metabolism (46, 47). CycADS facilitates the collection and management of information from genomic data and various protein annotation methods in an SQL database. The annotation methods aggregated by CycADS are the ones coming from Blast2GO (48, 49), InterProScan (50), KAAS-KEGG (51), and PRIAM (52) pipelines. A quality score (number of evidence) is generated for each annotation, and the predictions of all methods are displayed on the genes and proteins pages of the Cyc database, allowing users to assess the quality of the annotation of enzymes in the network. The data

**TABLE 1** Organisms contained in the ArtSymbioCyc collection[a]

| Organisms (hosts and associated bacteria) | Reference(s) |
|---|---|
| *Acyrthosiphon pisum* AL4f | (19) |
| *Buchnera aphidicola* APS | (20) |
| *Candidatus Hamiltonella defensa* T5A | (21) |
| *Sipha maydis* Midelt | (16) |
| *Buchnera aphidicola* Sm Midelt | (17) |
| *Serratia symbiotica* Sm Midelt | (17) |
| *Cinara cedri* | (22) |
| *Buchnera aphidicola* BCc | (23, 24) |
| *Serratia symbiotica* Cc | (25) |
| *Cimex lectularius* Harlan | (26) |
| *Wolbachia* sp. | (27) |
| *Drosophila melanogaster* | (28) |
| *Lactiplantibacillus plantarum plantarum* NC8 | (29) |
| *Acetobacter pomorum* WJL DM001 | (30) |
| *Wolbachia* sp. | (31) |
| *Glossina morsitans* Yale | (32) |
| *Sodalis glossinidius morsitan* | (33) |
| *Wigglesworthia glossinidia* (Yale colony) | (34) |
| *Bemisia tabaci* MEAM1 | (35) |
| *Candidatus* Portiera aleyrodidarum MEAM | (35) |
| *Hamiltonella defensa* MEAM1 | (36) |
| *Rickettsia* sp. MEAM1 | (37) |
| *Bemisia tabaci* MED | (38) |
| *Candidatus* Portiera aleyrodidarum BT-QVLC | (39) |
| *Hamiltonella defensa* MED | (40) |
| *Candidatus* Cardinium hertigii | (41) |
| *Wolbachia* sp. | (42) |
| *Pediculus humanus* corporis USDA | (43) |
| *Candidatus* Riesia pediculicola USDA | (43) |
| *Sitophilus oryzae* Bouriz | (44) |
| *Candidatus* Sodalis pierantonius SOE | (45) |

[a]See Table S1 for more information about the different databases, genome accession numbers, and specific features of the corresponding metabolic networks.

collected in CycADS are then formatted to generate *ad hoc* files ("Path-o-logic files") used by the Pathway Tools software (13) to produce a BioCyc-type-enriched metabolic database (12). Figure 1 schematically represents the complete process of annotating, storing, and organizing information, including the various CycADS processes that enable the production of BioCyC databases (PGDB) at the organism level or in a holobiont annotation process so-called "multi-organisms" in the Pathway Tools process. ArtSymbioCyc can be used to carry out broad comparative analyses between associated or non-associated organisms.

## ArtSymbioCyc to build high-quality genome-scale metabolic networks

To compare our reconstruction process with arthropods and symbiont genome-scale metabolic models (GSMMs) developed for flux balance analysis (FBA) (53, 54), we retrieved the Enzyme Commission (EC) numbers of 11 GSMMs (Table S2), recovering most of the up-to-date models available in the literature for these organisms. We compared their EC numbers with the ones obtained from our metabolic networks from ArtSymbioCyc for the same organisms. EC numbers were chosen as a comparative metric, as the reaction or metabolite identifiers are not homogeneous between the different GSMMs or between a GSMM and ArtSymbioCyc. EC numbers are identifiers for enzymatic activity, and they can be used as indicators of the coverage of metabolic activities in the different reconstructions.

We plotted the overlaps or specificities of EC numbers using area-proportional Venn diagrams (Fig. S1). For 10 out of the 11 GSMMs, we found that GSMMs are subsets of ArtSymbioCyc: almost all the EC numbers of the GSMMs are found in ArtSymbioCyc networks, but an important fraction of EC numbers is present only in ArtSymbioCyc.

The only GSMM with coverage of metabolic capacities close to ArtSymbioCyc is the work of Cesur et al. (55) on *D. melanogaster*. The authors took advantage of a large amount of information available on HMR2 (human metabolic reaction database 2) and adapted it to *D. melanogaster* using orthologies between the human and the *D. melanogaster* genomes available in the FlyBase database (56). They also integrated metabolic reconstruction based on the MetaCyc database (based on the same reconstruction tools as ArtSymbioCyc) (15). This shows that the BioCyc-based reconstruction process could be systematically integrated into all GSMMs to extend the metabolic coverage of each organism as far as possible, and our database is a valuable tool for this purpose. However, it would require very intensive additional work to achieve the quality and coverage of the *D. melanogaster* model in the non-model organisms we study here.

For *A. pisum* (the pea aphid), we manually examined the EC numbers that do not overlap (i.e., specific to either Blow's GSMM [57] or ArtSymbioCyc). Of the 14 EC numbers that are GSMM specific, 8 of them are obsolete EC numbers that have been reclassified in ArtSymbioCyc reconstruction. For the six remaining cases, we found that a very close enzymatic activity was found in ArtSymbioCyc for the associated genes, such as for D-aspartate oxidase (written as 1.4.3.1 in the GSMM vs 1.4.3.15 in ArtSymbioCyc) or phosphopantothenate-cysteine ligase (written as 6.3.2.5 in the GSMM vs 6.3.2.51 in ArtSymbioCyc). For this latter enzyme, we note that the ArtSymbioCyc EC number might be more accurate, as it is described in the KEGG database (58) as the eukaryotic version of the enzyme. For the last two cases, we estimated that a manual investigation is required to decide which annotation is the most appropriate: the gene LOC100168402 is identified as encoding for an enzyme that acts on GTP in the GSMM, whereas it is identified as an enzyme that acts on dUTP and dCTP in ArtSymbioCyc. Conversely, we examined the EC numbers of *A. pisum* that are specific to ArtSymbioCyc. Some of them are related to the degradation of key metabolites, such as amino acids and nucleotides. Also, many of them are related to secondary metabolism, including biosynthesis and degradation of sphingolipids (ceramide), carotenoids (retinoate and neurosporene), and catecholamine neurotransmitters (dopamine and norepinephrine). These families of metabolites are known to be of great importance for aphids (59) or more broadly for insects (60–62). Overall, the additional metabolic capacities provided by ArtSymbioCyc

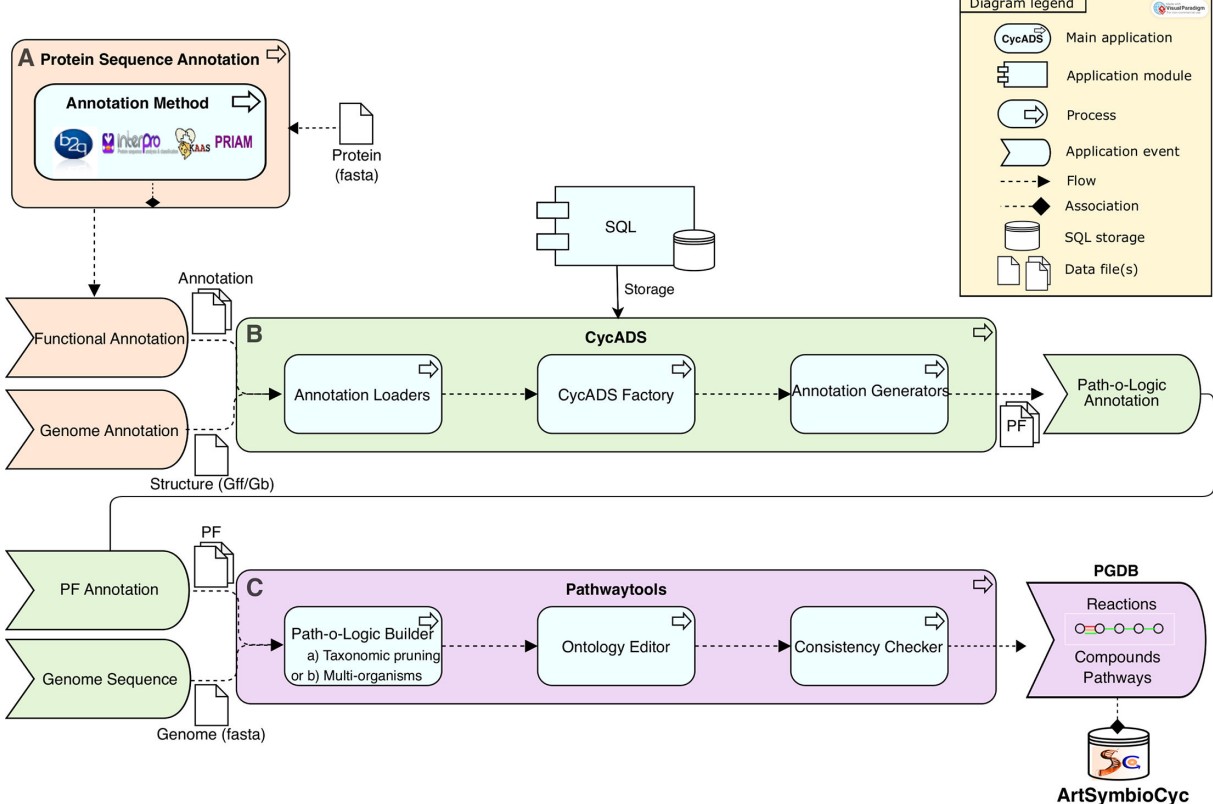

**FIG 1** The three-step pipeline for metabolic network reconstructions of ArtSymbioCyc. (A) Functional annotations of the protein data set (protein fasta file input) are predicted by several annotation methods (i.e., Blast2GO, Interproscan, KAAS, and Priam). (B) CycADS collects these annotation files together with the gff/gb genome structural annotation files (annotation loaders process). CycADS creates its own internal representation of information (CycADS factory process), enabling it to be stored in an SQL database and to generate structured annotations for Path-o-Logic tools (annotation generators process). (C) Pathway tools build the PGDB from the Path-o-Logic File (PF) information and the genome sequence based on MetaCyc, using (i) pruning of the reconstructed pathways for a simple organism or (ii) without taxonomic pruning for a multi-organism reconstruction mixing prokaryotic and eukaryotic pathways (Path-o-Logic Builder process). The ontology editor process is used to add useful metadata such as sequence source information, associated publications, external links description, etc., and finally, a consistency checker process is applied to each PGDB. An example of representation, termed glyph, can also be found in the web interface. It represents the sequence of reactions (lines) undergone by compounds (rounds) in a pathway present in the PGDB. These PGDBs are stored in an SQL database. They can be accessed through the ArtSymbioCyc interface.

show that GSMMs lack secondary metabolism and some degradation processes. These absences in GSMMs are not expected to be detrimental for FBA as this latter approach aims to (semi-)quantitatively model growth processes based on the main biomass components (e.g., DNA, RNA, proteins…) (53). A GSMM that is too large would also increase the risk of modeling artifacts that can occur in FBA and would require extensive manual curation (54). However, for a more qualitative and exploratory analysis of arthropod/symbiont metabolisms, crucial pathways for organisms' adaptation or development are missed if we only focus on incomplete GSMMs tailored for FBA.

## ArtSymbioCyc to ease and speed up metabolic network comparisons and analyses of host symbionts complementation

Many studies have been devoted to comparing symbiotic systems in order to decipher nutritional complementation (63–67) or to study genome evolution and the selective pressures exerted on the genomes of associated partners (68–70). For the first time, the ArtSymbioCyc collection and its BioCyc interface offer the community powerful tools capable of automatically carrying out a very large number of comparative analyses with high-quality graphical outputs (71). As an example, Table S3 compares the main biosynthetic pathways between the three aphid symbiotic systems *A. pisum*, *C. cedri*,

and *S. maydis*, and Fig. S2 compares the central metabolic pathways of these three host species. It should be noted, however, that human expertise is still required to validate or complete certain data (e.g., arginine biosynthesis in Table S3).

## A case study: the tripartite cooperation between *S. maydis* and its nutritional symbionts

The newly sequenced *S. maydis* genome (16) encodes 2,523 enzymes involved in 273 metabolic pathways. As a comparison, the reference *A. pisum* genome (AL4f, 23 March 2018) encodes 3,897 enzymes involved in 289 metabolic pathways. The two aphid holobionts are expected to differ in several aspects (Table S1), as the pea aphid contains only the primary symbiont *B. aphidicola,* whereas, in *S. maydis*, *B. aphidicola* with a reduced genome of 0.46 Mbp cooperates with the more recently acquired co-obligate symbiont *S. symbiotica* in several metabolic pathways. The most important differences between the two metabolic networks are summarized in Fig. 2. Overall, *S. maydis* appears as a metabolic subset of *A. pisum* and has restricted catabolic capacities as it is unable to assimilate certain carbohydrates, inorganic sulfur, and choline.

## Amino acid metabolism in *S. maydis*

From the ArtSymbioCyc interface, users can track the fate of compounds at the end (so-called dead-end products) or at the beginning (so-called precursors) of a metabolic pathway. Thus, we have been able to reconstruct the integrated metabolic networks for the biosynthesis of amino acids shared between *S. maydis* and its symbionts. Amino acid production is partitioned between *B. aphidicola* and *S. maydis*, which are therefore dependent on each other, while *S. symbiotica* reveals to be a sink for these important compounds (Fig. 3). With the exception of glycine, which is a metabolic hub, both nutritional symbionts have lost most of their ability to produce the non-essential amino acids that are either produced by the host (glutamate, aspartate, serine, proline, and tyrosine), or found in abundance in phloem sap (glutamine and asparagine [72]). Regarding the essential amino acids, *S. symbiotica* has conserved two incomplete pathways, which are partially redundant with those present in *B. aphidicola*, suggesting that the co-obligate symbiont could boost the system by producing chorismate for the three aromatic amino acids phenylalanine, tyrosine, tryptophan, and meso-diaminopimelate for lysine. *S. symbiotica* is therefore dependent on *B. aphidicola* and/or the aphid to produce all the essential amino acids.

B. aphidicola* has the capability to produce histidine, threonine, tryptophan, and phenylalanine, although the last transamination step from phenylpyruvate to phenylalanine is performed only in the host compartment, which is likely to prevent the symbiont to produce this amino acid in a selfish manner, as reported for the pea aphid-*Buchnera* system (46). Again, as in the case of the pea aphid, tyrosine is produced solely by *S. maydis* from phenylalanine, itself synthesized from phenylpyruvate supplied by *B. aphidicola*. A reversible interconversion between glycine and threonine is encoded in the aphid genome (via L-threonine aldolase), but this is more likely to be a threonine salvage reaction than a biosynthetic one (73). *B. aphidicola* can synthesize the three branched amino acids (valine, isoleucine, and leucine) up to the final transamination step, the latter being performed by the aphid. The lysine biosynthesis pathway is complete in *B. aphidicola*, while the last step is lacking in *S. symbiotica*.

The complete biosynthetic pathways of cysteine and methionine from central precursors require the transformation of assimilated sulfate into sulfite for the incorporation of sulfur into the molecules. In *B. aphidicola* and *S. symbiotica*, these pathways are not functional. Similarly, *S. maydis* has lost the ability to produce sulfite from sulfate. Cysteine biosynthesis from methionine (via homocysteine) is possible in *S. maydis* but not in *B. aphidicola* and *S. symbiotica*. Cysteine is probably absent or present in very low quantities in the phloem of Poaceae (42, 72), while S-methyl-L-methionine (SMM) has been shown to be the main form of methionine circulating in wheat phloem (74). As *S. maydis* can demethylate SMM using the enzyme homocysteine S-methyltransferase,

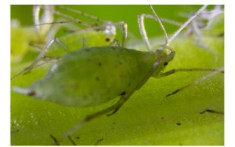 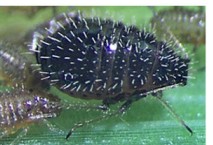

| | | *A. pisum* | *S. maydis* |
|---|---|---|---|
| **Biosyntheses** — amino acids | Arg | ✓ *Ba* | ↻ Aphid / *Ba* |
| | Sulfurated amino acid | ✓ *Ba* | 🌱 |
| | Chorismate | ✓ *Ba* | ✓ *Ba* ✓ *Ss* |
| | DAP | ✓ *Ba* | ✓ *Ba* ✓ *Ss* |
| **Biosyntheses** — vitamins | Coenzyme A | ↻ 🌱 / *Ba* | ↻ 🌱 / *Ss* |
| | B1 | 🌱 | ✓ Ss |
| | B2 | ✓ *Ba* | ✓ *Ss* |
| | FRDP (heme O precursor) | ✓ Aphid ✓ *Ba* | ✓ Aphid ✓ *Ss* |
| | NAD from NADP | ✓ Aphid | X |
| **Assimilation** — sulfur | Sulfate | ✓ *Ba* | X |
| | Sulfite | ✓ Aphid ✓ *Ba* | X |
| **Assimilation** — choline | Phosphatidylcholine hydrolysis | ✓ Aphid | X |
| | Glycine betaine production | ✓ Aphid | X |
| **Assimilation** — carbo-hydrate | Mannose | ✓ Aphid | X |
| | Mannitol | ✓ Aphid | X |
| | Mellobiose | ✓ Aphid | X |

✓ Achievable

↻ Codependence ( 🌱 means codependence between diet and one organism)

X Unachievable, expected to be absent/undone in holobiont metabolism

🌱 Unachievable, expected to be compensated by a supply from aphid diet

**FIG 2** Main differences between the metabolisms of *A. pisum* and *S. maydis* holobionts. Differences between the two *B. aphidicola* strains that have no direct impact on their respective host's metabolism are not shown. The organism achieving or co-achieving the metabolic process is referred to as aphid for *A. pisum* or *S. maydis*, Ba for *B. aphidicola*, or Ss for *S. symbiotica*. DAP: L,L-diaminopimelate; FRDP: farnesyl diphosphate.

SMM is the best candidate (rather than methionine) to enter the system and be used as a precursor for methionine and cysteine biosynthesis that are distributed to the symbiotic bacterial partners. This dependence on the organic sulfur source in the phloem appears to be a rather specific feature of this holobiont, as *A. pisum* holobiont achieves both inorganic sulfur assimilation and sulfur-containing amino acid biosynthesis (Fig. 2). This could be linked to the abundance of SMM in the phloem of Poaceae such as wheat (74).

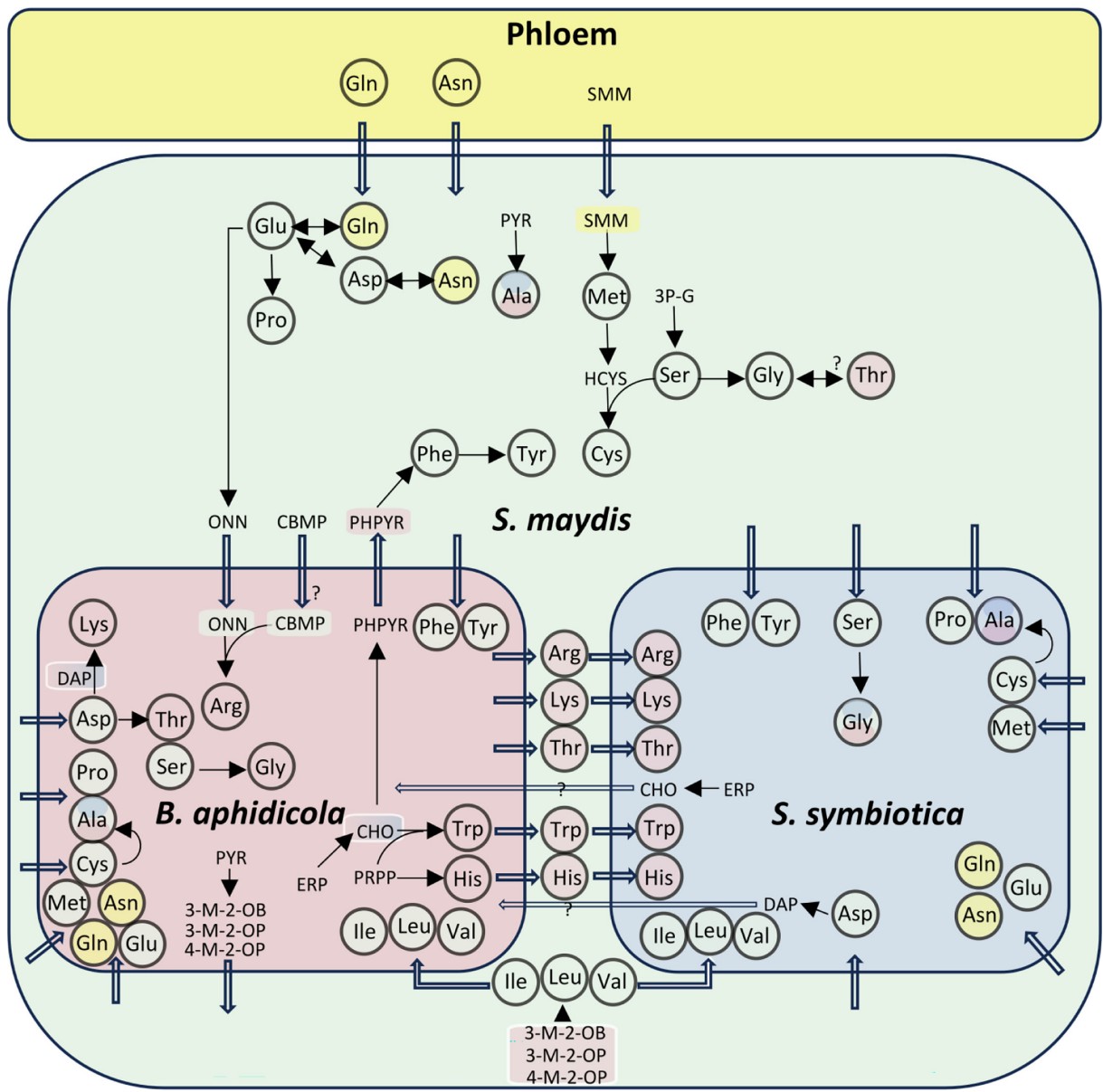

**FIG 3** The integrated metabolic network for amino acid biosynthesis of *S. maydis* holobiont. Final products (amino acids) are framed by a circle, and the unframed compounds represent the precursors. Amino acids and precursors are colored according to the compartment where they can be biosynthesized (*S. maydis*, *B. aphidicola*, or *S. symbiotica* cells). Some are hence bicolored or tricolored. Linear arrows represent a biosynthetic pathway, and thick empty arrows represent transport reactions. Question marks correspond to the hypotheses raised by the genomic inference we carried out for the present work. 3-M-2-OB: 3-methyl-2-oxobutanoate; 3-M-2-OP: 3-methyl-2-oxopentanoate; 3 P-G: 3-P-D-glycerate; 4-M-2-OP: 4-methyl-2-oxopentanoate; CBMP: carbamoyl phosphate; CHO: chorismate; DAP: L,L-diaminopimelate; ERP: D-erythrose 4-phosphate; HCYS: homocysteine; ONN: ornithine; PHPYR: phenylpyruvate; PRPP: 5-phospho-α-D-ribose 1-diphosphate; PYR: pyruvate; SMM: S-methyl-L-methionine.

In *B. aphidicola*, only the last part of the arginine pathway, which consists of the synthesis of arginine from ornithine, is conserved. Furthermore, Renoz et al. (17) reported that only one gene (*carA*) of the two (*carA* and *carB*) required for the production of carbamoyl-P (CBMP), a required co-substrate of the pathway, is present in *B. aphidicola* symbiont of *S. maydis*. Therefore, we hypothesize that another unidentified gene replaces *carB*, or CBMP is produced by *S. maydis* and supplied to *B. aphidicola* for use in the subsequent steps. In *S. symbiotica*, the pathway is completely lacking. *S. maydis* is able to synthesize ornithine from glutamate and encodes the complete enzymatic complex for CBMP production but lacks the ability to perform the final steps in the

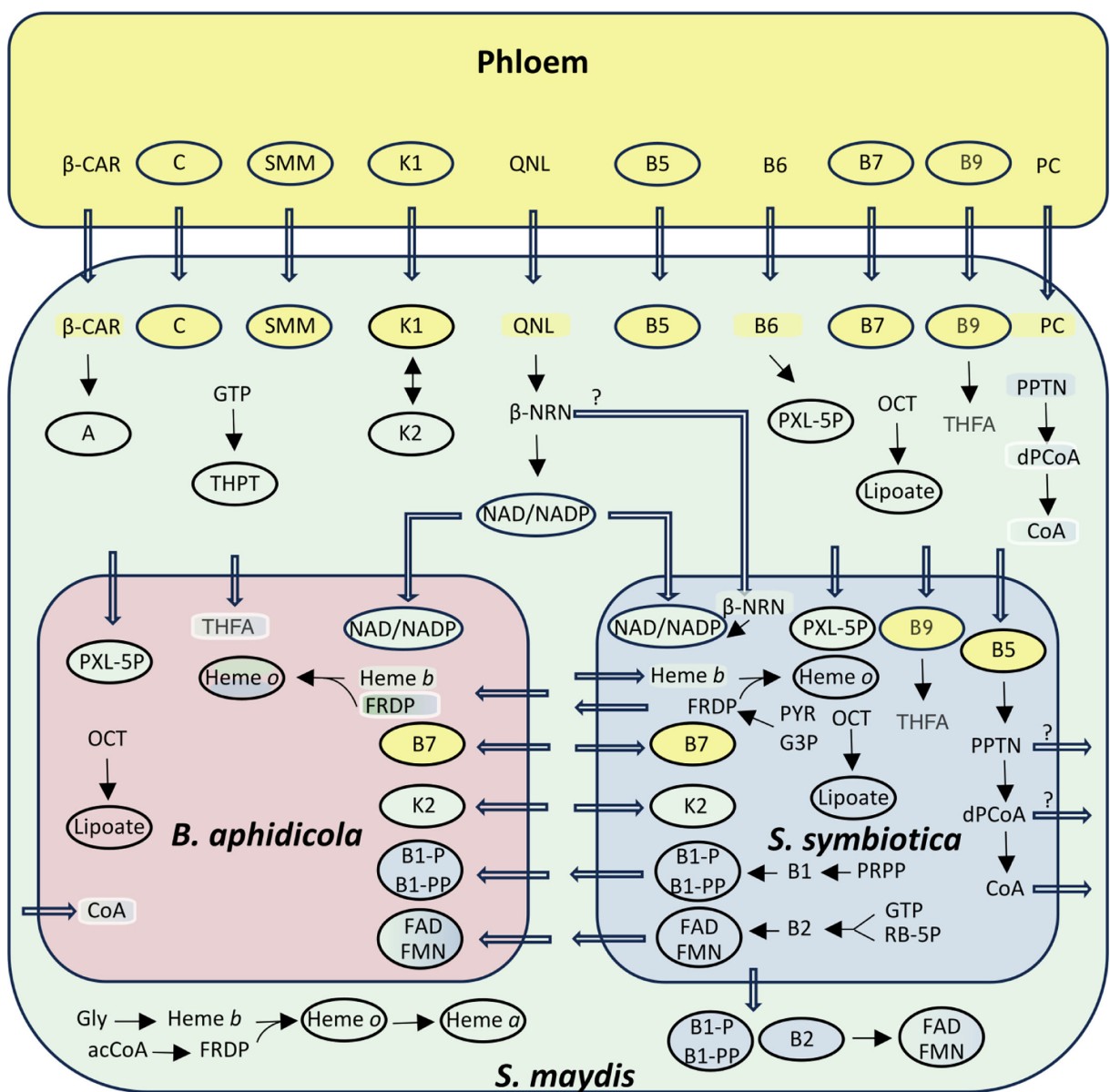

**FIG 4** The integrated metabolic network for the biosynthesis of vitamins of *S. maydis* holobiont. Final products (vitamins) are framed by a circle; unframed compounds represent precursors. Vitamins and precursors are colored according to the compartment where they can be biosynthesized (*S. maydis*, *B. aphidicola*, or *S. symbiotica* cells). Some are bicolored. Linear arrows represent a biosynthetic pathway, and thick empty arrows represent transport reactions. acCoA: acetyl coenzyme-A; β-CAR: β-carotene; β-NRN: β-nicotinate D-ribonucleotide; dPCoA: dephosphocoenzyme A; FAD: flavin adenine dinucleotide; FMN: flavin mononucleotide; FRDP: farnesyl diphosphate; G3P: glyceraldehyde 3-phosphate; GLY: glycine; PC: phosphatidyl choline; PPTN: phosphopantotheine; PRPP: 5-phosphoribosyl diphosphate; PXL-5P: pyridoxal 5-phosphate; PYR: pyruvate; QLN: quinolinate; RB-5P: ribulose 5-phosphate; THFA: tetrahydrofolate; THPT: tetrahydrobiopterin.

pathway leading to arginine production. This makes *S. maydis* and *B. aphidicola* co-dependent for arginine synthesis.

## Vitamin metabolism in *S. maydis*

In contrast to what we observed for amino acid biosynthesis, a small number of shared pathways between *S. maydis* and its nutritional symbionts have been found for vitamin biosynthesis (Fig. 4). The symbiotic system appears to rely heavily on the direct supply of vitamins from the plant for eukaryote-specific vitamins such as SMM (vitamin U), which

is also essential for methionine and cysteine biosynthesis (see above), ascorbate (vitamin C), and beta-carotene, the latter being converted by the aphid into retinol (vitamin A). The same observation applies to other more generalist vitamins that the host can absorb from the phloem sap and eventually distribute to its bacterial partners, which have lost the corresponding biosynthetic pathways: pantothenate (vitamin B5), biotin (vitamin B7), pyridoxine (vitamin B6), phylloquinone (vitamin K1), that can be converted by the aphid into menaquinone (vitamin K2), folate polyglutamate (the circulating form of B9 vitamin), that the host and *S. symbiotica* are able to convert into tetrahydrofolate.

Pantothenate (vitamin B5) is an important compound for insects and bacteria, notably for the production of the essential coenzyme A (CoA). However, *S. maydis* is unable to synthesize CoA from pantothenate and must export the latter to *S. symbiotica*, which has conserved the capability to produce CoA. This genomic analysis does not rule out the possibility that the bacterium returns intermediate compounds such as 4P-pantheine or dephospho-CoA to the host, as *S. maydis* encodes in its genome the final steps of the pathway from these precursors (Fig. 4). Conversely, *B. aphidicola* has completely lost the ability to produce CoA and must import it directly from *S. symbiotica* or *S. maydis*.

The case of the biotin (vitamin B7) biosynthesis pathway (Fig. 5) is puzzling. Indeed, the first part of the pathway, linked to fatty acid biosynthesis, is conserved in *B. aphidicola* and is missing in *S. symbiotica*. The final part of the pathway was considered, in a former analysis (17), as shared between *S. symbiotica*, which has conserved

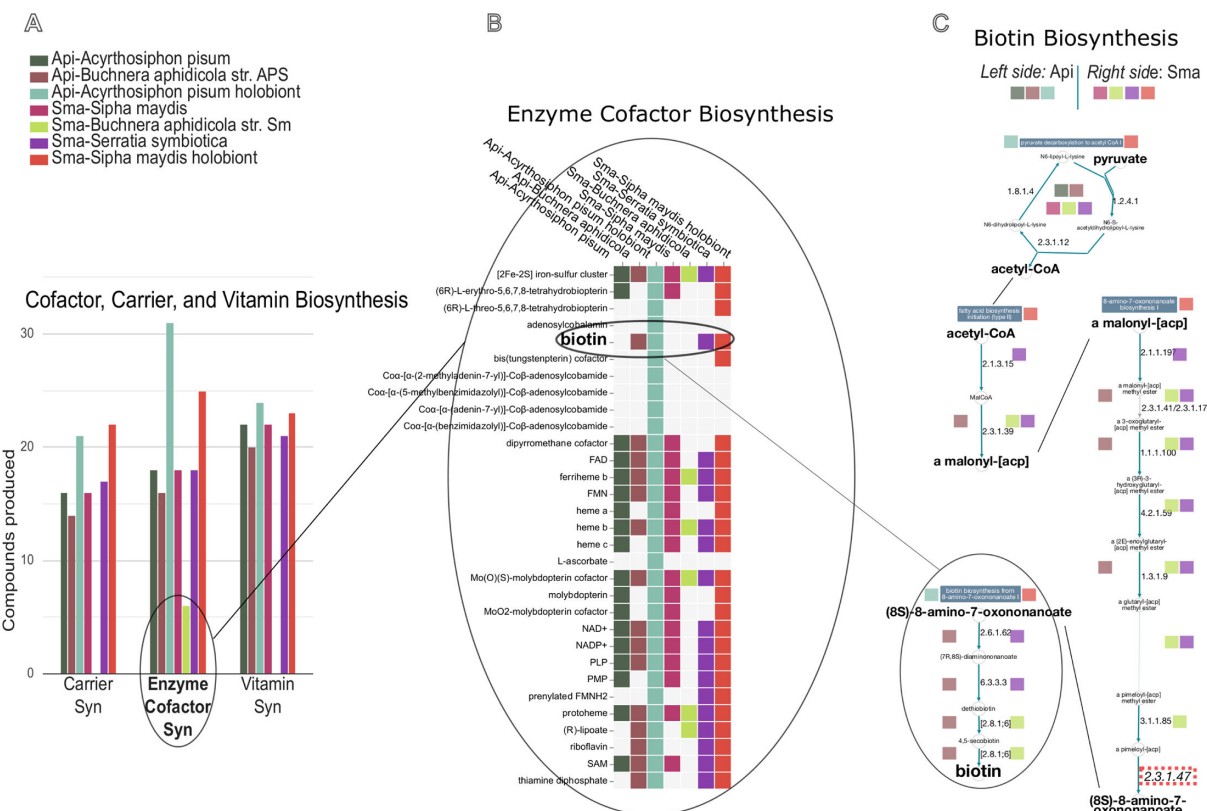

**FIG 5** Using the ArtSymbioCyc interface to compare biotin biosynthesis in *S. maydis* and *A. pisum* holobionts. See the color code in the legend to visualize the two compared holobionts *A. pisum* (left side) and *S. maydis* (right side). (A) Comparison of the number of compounds produced in different metabolic categories, including "synthesis of enzymatic cofactors." Clicking on the diagram gives access to synthetic square representations of the different pathways in this class of metabolism (B), including that of biotin. Then, with another click, you can access the details of the pathway, i.e., enzymes, reactions, and compounds within so-called pathway collages (C) to identify the overall capacities of the holobiont and its associated partners. In this biotin-specific analysis, we can see that complete biosynthesis is not possible since any of the holobiont partners can carry out step 2.3.1.47 (8-amino-7-oxononanoate synthase), visualized by the red dotted rectangle (bottom right).

the enzymes ensuring the first two steps, and *B. aphidicola*, which has conserved the enzymes ensuring the last two steps. The availability of the insect host genome shed new light on this pathway, showing that the 8-amino-7-oxononanoate synthase (EC 2.3.1.47), which links the two portions of the pathway, is absent not only from the symbiotic bacteria genomes but also from the *S. maydis* genome. Consequently, our new analysis suggests that biotin cannot be produced by the *S. maydis* symbiotic system (Fig. 5), and this vitamin is entirely supplied by the phloem sap as we propose in Fig. 4.

Nicotinamide (vitamin B3) is essential for the production of NAD and its derivatives. Complete NAD biosynthesis is typically of bacterial origin, but *B. aphidicola* and *S. symbiotica* have lost the corresponding genes. However, *S. maydis* can produce NAD from late intermediates such as quinolate or β-nicotinate D-ribonucleotide, which may be present in the phloem. Consequently, *S. maydis* can support *S. symbiotica* directly for NAD or for β-nicotinate D-ribonucleotide, as the end of the pathway is conserved in the bacterium. It should be noted that *B. aphidicola* must import NAD and NADP from its partners, as this symbiont has lost the enzymes needed to switch from one compound to the other.

The biosynthesis of thiamine (vitamin B1) and riboflavin (vitamin B2) is fully ensured by *S. symbiotica*, which encodes the entire pathway, from 5-phospho-α-D-ribose 1-diphosphate for B1 and from GTP and D-ribulose-5P for B2. It should be noted that, once again, *B. aphidicola* has lost the ability to produce active forms of the vitamins B1 and B2 (thiamine-P, thiamine-PP, flavin mononucleotide, and flavin adenine dinucleotide) and must obtain them from *S. symbiotica*. *S. maydis* can produce heme b from Gly and farnesyl diphosphate (FRDP) from acetyl-CoA, both required for heme O incorporation. *S. symbiotica* only needs to import heme b since the bacterium can synthesize FRDP from pyruvate and glyceraldehyde-3P, whereas *B. aphidicola* is totally dependent on the insect or on *S. symbiotica* for these biosynthetic pathways.

Lipoic acid is a cofactor for at least five enzymes, two of which belong to the citric acid cycle. The two enzymes required for lipoic acid biosynthesis are present in all three partners starting from octanoate, one of the very few cofactors that can be synthesized by *B. aphidicola*.

Tetrahydrobiopterin is an essential enzymatic cofactor (e.g., for the synthesis of aromatic amino acids), which is required in insects but not in bacteria. The tetrahydrobiopterin biosynthetic pathway is complete starting from GTP in *S. maydis*.

Although not a vitamin *per se*, phosphatidylcholine is supposed to be an essential membrane compound for the aphid, but its biosynthesis is not possible by any of the three associated members and must come from the plant (Fig. 4). The presence of lipids and phosphatidylcholine has been found in the phloem of plants of several families (75, 76), but not necessarily in Poaceae, as phloem lipids have been little studied to date and perhaps not at a concentration allowing their use for membrane structure. *B. aphidicola* and *S. symbiotica* may not require phosphatidylcholine, as has been documented for most bacteria that tend towards avirulence (77).

Finally, choline (not shown in Fig. 4) cannot be recovered from phosphatidylcholine in *S. maydis*. Moreover, the aphid, as well as its symbionts, has lost the capability to produce the osmoprotective glycine betaine from choline degradation, unlike *A. pisum,* which has retained both two metabolic capabilities (Fig. 2). Although glycine betaine transport by the phloem within the plant remains poorly understood, it is abundant in Poaceae (78), and this plant source might directly feed the holobiont (thus relieving it of choline). It is also possible that bacterial partners do not need glycine betaine by producing analogous osmoprotectants, such as ethanolamine or glycerol derivatives, as it has been suggested for the commensal microbiota of *D. melanogaster* (79).

## Concluding remarks

We show that ArtSymbioCyc is a unique resource for exploring and comparing the genome-wide metabolic networks of associated partners. It is also a tool for detecting critical nutritional interactions and thus for better understanding the adaptive potential

of symbiotic organisms. ArtSymbioCyc is also a powerful tool for producing high-quality genome-scale metabolic networks with homogeneous annotations predicted by the CysADS system.

By focusing on amino acid and vitamin metabolism in the *S. maydis* holobiont, we showed that the three symbiotic partners evolved together to specifically exploit the phloem of Poaceae. For example, phloem SMM is probably the only source of sulfur for cysteine and methionine biosynthesis, and most vitamins can only be synthesized from specific plant precursors. Throughout their evolutionary history, the genomes of *B. aphidicola* and *S. symbiotica* have greatly reduced their repertoire of genes for vitamin and amino acid biosynthesis and have become highly dependent on their aphid host. Nevertheless, it is remarkable that the metabolisms are still highly intertwined, with obligatory mutual complementations between the three partners, as it has been shown in other aphids and more widely in other hemipterans (11, 67). This strong connectivity, probably selected during evolution to avoid the selfish behavior of one of the partners, seals their common destiny within the plural structure that is the holobiont (3). It should be noted, however, that symbiont exchanges are still possible in highly integrated symbioses, with new symbionts potentially contributing new genes and enabling their host to change niche (11) as well as that symbiont selfish dynamics are sometimes observed (64). Thus, the holobiont concept does not presuppose that the evolutionary interests of the associated partners are convergent, which has even led some authors to reject the term itself (80). In view of the importance of nutritional symbioses in the adaptation of a wide range of insects to their environment, including some of the most serious pests, the ArtSymbioCyc database is a valuable tool to pursue this line of investigation, notably by including more holobionts in the future.

## MATERIALS AND METHODS

### Database collection implementation

The 10 arthropod holobionts included in ArtSymbioCyc at the time of writing have been added mostly based on the previous studies carried out by the authors (Table S1). All the symbiotic organisms in ArthropodaCyc, the largest public database of arthropod metabolisms (81), will gradually be integrated with their corresponding endosymbionts into ArtSymbiocyc, but other organisms, as well as specific partial holobionts (i.e., *A. pisum—Buchnera* only), can be included on request.

The architecture and the various steps of genome annotation and metabolic network reconstruction are presented in Fig. 1 in the results section above. To remove putative contaminant bacterial sequences from host genomes, the CycADS annotation system has been improved as described in Baa-Puyoulet et al. (81). Proteins identified as bacterial contaminants are not eliminated from the database but are excluded from the Pathway Tools reaction inference (i.e., from the metabolic network). A warning message appears on their corresponding gene page.

ArtSymbioCyc fully leverages the comprehensive BioCyc interface and its metabolism data analysis tools. Using the BioCyc online interface, users can perform several analyses with advanced query tools and robust web-based genomic data viewers. Additionally, ArtSymbioCyc allows data downloads in various formats for further analysis, using tools like Cytoscape (82) and MetExplore (83), or for integration into custom analysis software and pipelines.

## ACKNOWLEDGMENTS

We would like to thank Juan Perez-Limon for his help in developing some of the parameters of the CycADS annotation system that feeds the ArtSymbioCyc database. We also thank Aurélie Herbomez for secretarial assistance.

Research at BF2i was supported by the Institut National de la Recherche pour l'Agriculture, l'Alimentation et l'Environnement (INRAE), and the Institut National des

Sciences Appliquées de Lyon (INSA Lyon). This work was supported by the Agence Nationale de la Recherche (ANR) programs "Co-adaptations hôtes-microbiote: mécanismes et conséquences–Hmicmac" (ANR-16-CE02-0014) and "Fight Bedbug Infestations: guide insecticide treatments and develop alternative methods of control- FBI" (ANR number PrANR-21-CE35-0011).

P.B.-P. and H.C. conceived the project. P.B.-P., N.P., and S.P. developed the database and the associated pipelines. P.B.-P. and L.G. produced the figures. P.B.-P., L.G., N.P., F.R., S.P., F.C., and H.C. analyzed the *S. maydis* holobiont data, contributed to the drafting of the manuscript, and revised and approved the final version of the manuscript.

## AUTHOR AFFILIATIONS

[1]INRAE, INSA Lyon, BF2I, UMR203, Villeurbanne, France
[2]Biodiversity Research Centre, Earth and Life Institute, UCLouvain, Louvain-la-Neuve, Belgium

## AUTHOR ORCIDs

Patrice Baa-Puyoulet http://orcid.org/0000-0002-7033-7245
Léo Gerlin http://orcid.org/0000-0002-1398-2790
Nicolas Parisot http://orcid.org/0000-0001-5217-8415
Sergio Peignier http://orcid.org/0000-0002-9004-3033
François Renoz http://orcid.org/0000-0003-3716-4562
Federica Calevro http://orcid.org/0000-0001-7856-9617
Hubert Charles http://orcid.org/0000-0002-0809-4705

## FUNDING

| Funder | Grant(s) | Author(s) |
| --- | --- | --- |
| Agence Nationale de la Recherche | ANR-16-CE02-0014 | Patrice Baa-Puyoulet |
| | | Nicolas Parisot |
| | | Sergio Peignier |
| | | Federica Calevro |
| | | Hubert Charles |
| Agence Nationale de la Recherche | PrANR-21-CE35-0011 | Patrice Baa-Puyoulet |
| | | Nicolas Parisot |
| | | Federica Calevro |
| | | Hubert Charles |

## AUTHOR CONTRIBUTIONS

Patrice Baa-Puyoulet, Conceptualization, Data curation, Formal analysis, Investigation, Methodology, Project administration, Resources, Software, Supervision, Validation, Visualization, Writing – original draft, Writing – review and editing | Léo Gerlin, Data curation, Formal analysis, Resources, Software, Validation, Visualization, Writing – original draft, Writing – review and editing | Nicolas Parisot, Data curation, Methodology, Software, Validation, Visualization, Writing – original draft, Writing – review and editing | Sergio Peignier, Software, Validation, Writing – original draft, Writing – review and editing | François Renoz, Visualization, Writing – original draft, Writing – review and editing | Federica Calevro, Funding acquisition, Writing – original draft, Writing – review and editing | Hubert Charles, Conceptualization, Formal analysis, Funding acquisition, Investigation, Methodology, Project administration, Resources, Software, Supervision, Validation, Visualization, Writing – original draft, Writing – review and editing

## DATA AVAILABILITY

Metabolic network reconstructions were carried out using Pathway Tools 27.0 (12 April 2023). Annual updates of ArtSymbioCyc are carried out to search for and re-annotate new versions of genomes when necessary and to monitor the evolution of the Pathway Tools required for network reconstruction and the analysis interface. Metabolic network reconstructions and the resulting BioCyc metabolism databases are available in the ArtSymbioCyc collection database (https://artsymbiocyc.cycadsys.org/). The annotations, inferred reactions, and networks files of the genomes or proteomes have also been made available in a dedicated data set (84).

## ADDITIONAL FILES

The following material is available online.

### Supplemental Material

**Supplemental material (mSystems00140-25-s0001.pdf).** Figures S1 and S2; Tables S1 to S3.

### Open Peer Review

**PEER REVIEW HISTORY (review-history.pdf).** An accounting of the reviewer comments and feedback.

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
