## [Reviewer comments · mSystems]

ArtSymbioCyc, a metabolic network database collection dedicated to arthropod symbioses: a case study, the tripartite cooperation in *Sipha maydis*

Patrice Baa-Puyoulet, Leo Gerlin, Nicolas Parisot, Sergio Peignier, François Renoz, Federica Calevro, and Hubert Charles

Corresponding Author(s): Hubert Charles, Institut National des Sciences Appliquées de Lyon

Review Timeline:

Submission Date:

January 29, 2025

Accepted:

February 20, 2025

Editor: Jonathan Klassen

Reviewer(s): Disclosure of reviewer identity is with reference to reviewer comments included in decision letter(s). The following individuals involved in review of your submission have agreed to reveal their identity: Rosario Gil (Reviewer #1)

Transaction Report:

DOI: <https://doi.org/10.1128/msystems.00140-25>

Re: mSystems00140-25 (**ArtSymbioCyc, a metabolic network database collection dedicated to arthropod symbioses: a case study, the tripartite cooperation in *Sipha maydis***)

Dear Prof. Hubert Charles:

Your manuscript has been accepted, and I am forwarding it to the ASM production staff for publication. Your paper will first be checked to make sure all elements meet the technical requirements. ASM staff will contact you if anything needs to be revised before copyediting and production can begin. Otherwise, you will be notified when your proofs are ready to be viewed.

Sincerely,
Jonathan Klassen
Editor
mSystems

Reviewer #1 (Comments for the Author):

The authors have responded more than adequately to all my suggestions. I sincerely believe that the manuscript, which already had sufficient merit for scientific advancement once some minor problems had been resolved, has been considerably improved by the new structuring, the incorporation of new models and the assessment of the advantages of ArtSymbioCyc to facilitate the work of researchers in the area. The possibility of periodic updates, new incorporations and reconstructing partial holobionts on demand is very relevant. The new supplementary material is also very informative.

I would also like to point out that, as the authors say, I had not detected any previous comparable studies that had been neglected. However, the incorporation of new references seems adequate to give a broader view of the process that has led to the current knowledge of symbiont models as a holobiont for the study of their metabolic networks as a whole, now that whole host genomes of acceptable quality are being produced. At the end, I also considered important the recognition that manual curation would be also essential and facilitated by ArtSymbioCyc.